# Relationship between Objective and Subjective Fatigue Monitoring Tests in Professional Soccer

**DOI:** 10.3390/ijerph20021539

**Published:** 2023-01-14

**Authors:** João Lourenço, Élvio Rúbio Gouveia, Hugo Sarmento, Andreas Ihle, Tiago Ribeiro, Ricardo Henriques, Francisco Martins, Cíntia França, Ricardo Maia Ferreira, Luís Fernandes, Pedro Teques, Daniel Duarte

**Affiliations:** 1Research Center of the Polytechnic Institute of Maia (N2i), Maia Polytechnic Institute (IPMAIA), Castêlo da Maia, 4475-690 Maia, Portugal; 2Department of Physical Education and Sport, University of Madeira, 9020-105 Funchal, Portugal; 3LARSYS, Interactive Technologies Institute, 9020-105 Funchal, Portugal; 4Center for the Interdisciplinary Study of Gerontology and Vulnerability, University of Geneva, 1205 Geneva, Switzerland; 5Research Unit for Sport and Physical Activity (CIDAF), Faculty of Sport Sciences and Physical Education, University of Coimbra, 3004-504 Coimbra, Portugal; 6Department of Psychology, University of Geneva, 1205 Geneva, Switzerland; 7Swiss National Centre of Competence in Research LIVES—Overcoming Vulnerability: Life Course Perspectives, 1015 Lausanne, Switzerland; 8Faculty of Human Kinetics, University of Lisbon, 1499-002 Lisbon, Portugal; 9Marítimo da Madeira—Futebol, SAD, 9020-208 Funchal, Portugal; 10Research Center in Sports Sciences, Health Sciences, and Human Development (CIDESD), 5000-801 Vila Real, Portugal; 11Department of Physiotherapy, School of Health Technology of Coimbra (ESTeSC), Polytechnic Institute of Coimbra (IPC), São Martinho do Bispo, 3045-093 Coimbra, Portugal

**Keywords:** football, fatigue, monitorization, vertical jump, well-being questionnaire

## Abstract

Studying fatigue is challenging because it is influenced by physiological, psychological, and sociological states. Fatigue can be assessed objectively or subjectively, but the literature has difficulty understanding how an analytical test relates to a response via a questionnaire. Thus, the purpose of this study was to evaluate the relationships between objective fatigue variables (Squat Jump (SJ) and Countermovement Jump (CMJ)) measured on day-2 to the game and subjective fatigue (Rating Perceived Exertion (RPE) measured on day-3 to the game and Hooper Index (HI) measured on day-2). The sample comprised 32 professional football players from the First Portuguese League aged 25.86 ± 3.15 years. The Spearman correlations and regression analyses were used to study the relationships between the variables. The results showed statistically significant (*p* < 0.05) but small correlations (0.113–0.172) between several objective metrics and the subjective metrics evaluated. In addition, we found two weak models with statistical significance (*p* < 0.05) between the dependent objective variables (contact time, height, and elasticity index) and the HI ($R^{2}$ = 3.7%) and RPE ($R^{2}$ = 1.6%). Also, nine statistically significant (*p* < 0.05) but weak models were observed between the subjective dependent variables (HI and RPE) and contact time ($R^{2}$ = 1.8–2.7%), flight time ($R^{2}$ = 1.1–1.9%), height ($R^{2}$ = 1.2–2.3%), power ($R^{2}$ = 1.4%), pace ($R^{2}$ = 1.2–2.1%), and elasticity index ($R^{2}$ = 1.6%). In conclusion, objective and subjective fatigue-monitoring tests in professional soccer do not measure identical but rather complementary aspects of fatigue, and therefore, both need to be considered to gain a holistic perspective.

## 1. Introduction

In team sports, the main objective of the training process is to manage the stimuli that optimize player/team performance for competition. Thus, sports agents and coaches seek to optimize players’ fitness levels without them being negatively affected by fatigue [1]. Professional soccer players can play up to 50 games per season with one or two competitive matches per week. Past research has described losses in physiological and performance measures following high training load sessions and reported that negative effects could last for days [2]. Therefore, the coaching staff should carefully manage and periodize the players’ process according to competitive and training demands to avoid the detrimental effects of accumulated fatigue, which may lead to an increased risk of injury [3,4]. For this reason, it is crucial to use a monitoring system that balances training load, recovery status, and readiness for competition [1,5], leading to well-structured training plans.

To help control and monitor training, new technologies have emerged in the sports context. For example, through the global positioning system (GPS), it is possible to monitor the internal and external load as well as study the position of players on the field and their respective running speeds and distances covered [6,7]. However, for some clubs, financial capacity may inhibit the implementation of monitoring instruments, such as GPS, on all players. Based on this, other tools have been adopted for monitoring players and the manifestation of fatigue [8], including the application of subjective well-being questionnaires (stress, fatigue, pain, and sleep—Hopper Index (HI)) and the Rate of Perceived Exertion (RPE) related to training units. While the RPE is collected after training sessions, the HI is collected before, allowing the coaching staff to detect individual signs of pre-fatigue and, eventually, adjusting the scheduled training load [9]. Although they are subjective, these are easy field tests that do not require a deep financial investment and have frequently been used to monitor fatigue in soccer players [8].

Indeed, both the HI and RPE were reported to be associated with training load in professional soccer players [9,10]. However, assessing the subjective measures of training load and well-being may be less reliable than using objective data. Therefore, combining subjective and objective variables, such as a vertical jumping assessment, could provide a more realistic picture of players’ status [5]. According to the literature, vertical jumps have been one of the most reliable measures to quantify sports performance and fatigue related to training in professional players [11]. However, more than analyzing physical and well-being data in isolation, there is a need to contextualize the data to clearly understand the results [12]. The assessment of fatigue in an objective and subjective way and a better understanding of the relationships between these variables allows a better understanding of fatigue behavior. However, few studies still evaluate weekly training load with athlete well-being indices. Thus, this study aimed to investigate the biopsychosocial aspect of fatigue during a sports season by studying the associations between objective (vertical jumps) and subjective (RPE-session and HI questionnaires) fatigue tests. It was hypothesized that both the RPE and the HI scores would present a significant relationship with vertical jumping performance in professional soccer players.

## 2. Materials and Methods

### 2.1. Study Design and Sample Size

This study is a correlational research design composed of a sample of 32 professional players competing at the First Portuguese Soccer League over the 2020/2021 season. The data were collected throughout the sports season at the training sessions. The inclusion criteria were (i) players who had represented the club for the entire study time and (ii) players included in the professional team. Regarding the exclusion criteria, players were not considered if (i) they had acute and/or chronic injuries that prevented them from playing, (ii) they failed the RPE-session (MD-3) and HI-session (MD-2) questionnaires, and failed to perform the vertical jumps in MD-2. This study did not consider individual training, recovery, and rehabilitation sessions.

The players were informed of the study design and the benefits and consequences of their participation and freely signed an informed consent form. All procedures were approved by the Ethics Committee of the Faculty of Human Kinetics, University of Lisbon (CEIFMH, No. 34/2021) and followed the ethical standards of the Declaration of Helsinki for a study in humans.

### 2.2. RPE Sessions

This procedure took place approximately 30 min after the end of the training sessions on match day-3 (MD-3); each player attributed a general classification according to their perception of effort, where 0 corresponded to no fatigue and 10 to extreme fatigue (i.e., Borg Scale CR10) [13,14]. All the records were made through a mobile application provided by the club or through record sheets designed for that purpose (i.e., alternative plan/prevention tool) at the locker room entrance (regardless of the team’s training location).

### 2.3. Wellness Questionnaire (HOOPER)

This process took place up to 30 min before the start of the training session on game day-2 (MD-2); each player was assigned a rating on a scale, where 1 corresponded to low predisposition and 5 to high well-being for the different sets of well-being (i.e., sleep, stress, fatigue, and muscle pain) [15]. Afterward, the HI was calculated using the sum of all well-being sets. The information was collected using a procedure similar to the RPE assessment.

### 2.4. Vertical Jumps

The squat jump (SJ) and the countermovement jump (CMJ) were selected to evaluate the vertical jumps and took place before the start of the training session on game day-2 (MD-2). The protocol was composed of four jumps, and the inclusion of arm balances was not allowed. The recovery time between repetitions was established in a 1/6 ratio in which “1” corresponded to the execution duration of one repetition, and “6” was the duration of the recovery time [16]. In the case of the CMJ, the depth of the countermovement was self-regulated by the players, but in the flight phase, they were obliged to maintain the extension of the lower limbs. Concerning the SJ, the players placed themselves in a squatting position (approximately 90°). For all jumps, two attempts were allowed for familiarization. In the case that the movement was considered wrong during the execution, it was repeated. The Optojump Next (Microgate, Bolzano, Italy) system of analysis and measurement was used to perform the vertical jumps. Through the software of this instrument, it was possible to access the following variables: (i) jump height, which corresponds to a change in the height of the athlete’s center of mass; (ii) flight time; and (iii) maximum power [17]. Besides the aforementioned data, the elasticity index (EI)—the elastic energy used in pre-stretching during the CMJ—was also calculated with the formula IE = (CMJ − SJ) × 100/SJ. Finally, the reactive strength index (RSI), which represents the ability to use the stretching–shortening cycle, was also calculated with the formula RSI = Jump Height/Ground Contact Time [18].

### 2.5. Statistics Analysis

Descriptive statistics were presented as the standard deviation of the means. The relationships between the RPE, HI, and vertical jumps were evaluated using Spearman’s rho correlation [19]. The strength of the relationship was evaluated according to the following criteria: 0 to 0.20—Insignificant; 0.21 to 0.40—Weak; 0.41 to 0.60—Moderate; 0.61 to 0.80—Strong; and 0.81 to 1.00—Very strong. Third, multiple and simple linear regressions were conducted [20]. The $R^{2}$ values were interpreted as follows: $R^{2}$ < 2%—Very weak; 2% ≤ $R^{2}$ < 13%—Weak; 13% ≤ $R^{2}$ < 26%—Moderate; and $R^{2}$ ≥ 26%—Substantial. The data analysis hypotheses were checked. The confidence level was set at 95%. All analyses were performed using IBM SPSS Statistics 28.0 software (SPSS Inc., Chicago, IL, USA).

## 3. Results

The participants included 32 male soccer players aged 25.86 ± 3.15 years (body mass index 23.8 ± 1.46 kg/m^2^, level of experience 8.28 ± 3.13 years). They were from different nationalities with Brazilian players constituting the vast majority of this sample (~58%). Furthermore, it is noteworthy that this group of players was composed of players from all positions, particularly midfield players who represented the majority. A more detailed characterization of the professional soccer players is presented in Table 1.

From the analyzed data, regarding well-being, it was found that sleep quality obtained the highest values (3.75 ± 0.76), and in contrast, fatigue obtained the lowest values (3.24 ± 0.68). Furthermore, all the CMJ values were superior to the SJ values (except the average contact time). Table 2 shows, in more detail, the results of the well-being questionnaire, HI, RPE, CMJ, and SJ scores.

Table 3 shows the associations between the variables RPE, HI, and vertical jumps (CMJ and SJ). A weak positive and significant correlation value was recorded between the variables RPE and HI (−0.345; *p* < 0.001).

Insignificant correlation values and significances were found between the CMJ and HI variables (0.172–0.148; *p* < 0.01). The results also showed minor negative correlation values with a moderate level of significance between the following variables: Pace_mean_CMJ and HI, and Height_mean_CMJ and RPE (0.126–0.113; *p* < 0.01). Finally, insignificant positive correlation values with a low level of significance were recorded between the SJ and HI variables (0.126–0.113; *p* < 0.05). Similarly, insignificant negative correlation values with a low level of significance were recorded between the variables Tflight_mean_CMJ, Height_mean_CMJ, Power_mean_CMJ, and RPE.

A statistically significant model was evidenced for the dependent variable HI ($R^{2}$ = 3.7%; *p* < 0.001) in which the independent variables were the mean CMJ contact time and mean CMJ height. A statistically significant model was also found for the dependent variable RPE ($R^{2}$ = 1.6%; *p* = 0.009) in which the independent variable was the mean EI. Other models were tested but were not shown to be statistically significant. Table 4 shows, in more detail, all the statistically significant models found.

The results showed several statistically significant models for the dependent variables mean CMJ contact time, mean CMJ jump height, mean CMJ pace, and mean CMJ flight time in which the independent variable was HI ($R^{2}$ = 1.9%—$R^{2}$ = 2.7%). Several statistically significant models 
were identified for the dependent variables mean SJ contact time, mean SJ jump 
height, and mean SJ flight time in which the independent variable was HI ($R^{2}$ = 1.1% and $R^{2}$ = 1.8%). Multiple statistically significant models were identified for the dependent variables mean EI, mean CMJ jump power, mean CMJ jump height, mean CMJ pace, and mean CMJ flight time in which the independent variable was RPE ($R^{2}$ = 1.2% and $R^{2}$ = 1.4%). Other models were run but were not statistically significant. For further analysis, see Table 5.

## 4. Discussion

In professional soccer, the study of the relationship between psychometrics and physical and biological responses during the training process is of great importance for effectively managing the training load to prevent the negative effect of fatigue [21]. The results of the present study revealed a weak relationship between the two subjective tests (the HI and the RPE), which indicates that it is not always true that when an athlete is fatigued at the end of the training, they have little predisposition to train the next day. This can be supported by other studies that showed weak to moderate correlations when analyzing the same variables (HI: 0.164, *p* < 0.01; and RPE: 0.47, *p* < 0.01) [9]. Furthermore, a linear regression and correlation analysis showed a negative correlation between the RPE mean and the sum of all the well-being variables [22]. This corroborates a meta-analysis work, where a weak relationship between workload measured with the RPE and well-being was revealed [23]. This exalts the biopsychosocial aspect of fatigue, where it is understood that many factors besides the load influence the well-being scores. In fact, the RPE can be used as an indicator of the work done by players (external load) and the physiological stress imposed on them (internal load) [5]. On the other hand, the HI can be used as an indicator of overtraining and recovery [9].

As it is known, an intensification of the training load causes disorders, such as poor sleep quality, increased stress, fatigue, worsening of muscle pain, and a longer recovery [15,24]. In this sense, when the wellness questionnaire is analyzed in detail by recording the players’ average scores throughout the sports season, it is found that fatigue (3.24) and muscle pain (3.29) are the parameters that most affect athletic well-being. This finding was explored in another study that reported that fatigue and muscle pain increased throughout the season in relation to sleep quality and stress [25]. Similarly, another study highlighted a greater awareness of fatigue and muscle pain concerning the other components of well-being [9]. For these reasons, it is considered essential to have these components analyzed in more detail in a questionnaire to evaluate well-being.

During the analysis of the correlations between the subjective tests (HI and RPE) and the objective tests (SJ and CMJ), it was observed that the flight and height variables (Tflight_mean and Height_mean) in both jumps showed significant correlations with the subjective tests. This might indicate that both variables are more susceptible to fatigue. Something similar was seen in another study, where several components of the CMJ were analyzed along with their correlations with perceived training load and well-being values. They reported that flight time reported negative changes when the training load increased, and well-being decreased [14]. So, in these variables, a special look should be taken when studying athlete fatigue.

Interestingly, the HI showed higher correlations with the objective jump metrics compared to the RPE. On the other hand, the jumping components of the CMJ showed an overwhelming majority of correlations with the two subjective tests, where we can understand that the CMJ is more predominant for fatigue control than the SJ. This data reinforces the study of the type of jump with greater reliability to evaluate performance and neuromuscular status [26,27].

Regarding the linear regressions, the components of the CMJ stood out with the variables Tcont_CMJ_mean and Height_CMJ_mean being present in the statistically significant models. In addition, the HI was more sensitive to regressions than the RPE, since we found two explanatory models for the HI and only one for the RPE. This seems to suggest that fatigue is complex and biopsychosocial, since although there is a relationship between neurological ability measured with jumps and well-being, other contextual factors and intrapersonal behaviors can modulate well-being scores. Also, it is important to note that perceived training effort responses and well-being scores depend on the type of task the players perform.

Finally, we found 12 explanatory linear regression models of the vertical jump components studied, where it was found that, again, flight time proved to be the jump component most sensitive to fatigue. Furthermore, the CMJ is, again, when compared to the SJ, more susceptible to variation and fatigue control when compared to the subjective measures. Additionally, the overwhelming majority of the regressions of the vertical jump components, whether measured with the CMJ or the SJ, predicted greater change for the HI values than for the RPE. However, a moderate to strong correlation was observed between the players’ perceived fatigue and day-to-day change in high-intensity running (r = −0.51; *p* < 0.001). The slope of the regression model indicated that each increase of approximately 400-m in the distance led to an increase of 1 unit on the fatigue scale. In this sense, it appears that the RPE relates better to external load measures, such as running, than internal ones, such as the jump [28]. In a study conducted in elite Australian footballers, the authors reported that every additional 2.1 km of sprint distance covered during the pre-season worsened the players’ in-season well-being scores by approximately 1.2 points, which underlines the ability of the well-being assessment to be associated with objective measures and fatigue [29].

The connection between the CMJ and HI may be explained by the fact that muscle pain and fatigue compromise athlete performance [30]. Additionally, in soccer, the actions involve all phases of the stretch–shortening cycle (SSC) that can be measured with the CMJ and have an influence on speed, jumping ability, and change in direction capacity. Thus, understanding the SSC is important to delimit the ability to work at high intensities [31,32]. These actions being repeated with some frequency without sufficient recovery compromise force production at the SSC due to neuromuscular fatigue [33], where delayed onset muscle soreness (DOMS) and fatigue may be responsible for reducing performance.

## 5. Conclusions

Relationships between the objective and subjective variables were found, but they were too weak at predicting relevant changes. These results support using objective and subjective fatigue-monitoring tests as complementary aspects of fatigue, since they do not measure identically. While objective tests represent the physiological and physical aspects, subjective tests may represent psychological and sociological variables. Therefore, using both testing approaches may enhance a holistic player’s evaluation.

On the other hand, it was possible to verify that the HI was the subjective test most associated with the objective tests, while the CMJ was the most related to the subjective measures. Still referring to these metrics, height and flight time for the CMJ were the components that correlated the most and had the best results with the subjective tests. On the other hand, in the subjective context, the metrics found for the wellness questionnaire reveal that we should pay special attention to DOMS and the fatigue that the athlete may report.

As for the linear regression models explaining the HI and RPE, only two were found. On the other hand, 12 were found in the linear regression models describing the vertical jump components. In other words, it was possible to predict more changes in the objective tests through the subjective metrics than the opposite. The HI was more predictive of the objective tests, and the CMJ was more predictive of the subjective tests.

## 6. Limitations and Future Investigations

One of the limitations of this study is that only one senior professional team was evaluated. Throughout the season, some players became unavailable (injuries, leaving the club, personal reasons, etc.), making it difficult for the data assessment and evaluation. So, it would be interesting to use the same methodology in other professional clubs.

An interesting investigation, in the context of fatigue assessment, would be the use of the subjective HI test to study the regressions and correlations with the field objective control metrics, where we would try to understand on day-3 of the game, the impact of using GPS data (changes in direction, accelerations, decelerations, and distances traveled at high speed) in the predisposition for the next-day training, measuring the well-being at the beginning of day-2 to the game.

## Figures and Tables

**Table 1 ijerph-20-01539-t001:** Descriptive statistics for the sample characteristics.

	Mean ± SD	No. (%)
Age	25.86 ± 3.15	
Body mass index	23.8 ±1.46	
Years of experience	8.28 ± 3.13	
Players per Nationality		
Angola		1 (3.12%)
Argentina		1 (3.12%)
Brazil		11 (34.6%)
Brunei		1 (3.12%)
Cameroon		1 (3.12%)
Colombia		2 (6.2%)
Cyprus		1 (3.12%)
Democratic Republic of Congo		1 (3.12%)
France		2 (6.2%)
Iran		2 (6.2%)
Italy		1 (3.12%)
Mozambique		1 (3.12%)
Portugal		5 (15.6%)
Serbia		1 (3.12%)
Switzerland		1 (3.12%)
Sectorial position		
Goalkeepers		3 (9.37%)
Defenders		10 (31.25%)
Midfielders		13 (40.62%)
Forwards		6 (18.75%)

**Table 2 ijerph-20-01539-t002:** Descriptive statistics concerning the well-being, HI, subjective perception of effort, and explosive force of the participants.

	Mean ± SD
Sleep Quality	3.75 ± 0.76
Stress Level	3.61 ± 0.72
Muscle Pain	3.29 ± 0.73
Fatigue	3.24 ± 0.68
HI	13.90 ± 2.27
RPE	4.86 ± 1.73
Tcont_CMJ	4.06 ± 1.42
Tflight_CMJ	0.55 ± 0.04
Height_CMJ	37.75 ± 4.55
Power_CMJ	15.34 ± 1.34
Pace_CMJ	0.26 ± 0.16
RSI_CMJ	0.10 ± 0.0
Tcont_SJ	5.12 ± 1.62
Tflight_SJ	0.53 ± 0.03
Height_SJ	35.77 ± 4.22
Power_SJ	14.37 ± 1.14
Pace_SJ	0.20 ± 0.11
RSI_SJ	0.07 ± 0.02
IE	5.90 ± 9.36

± = standard deviation; CMJ = Countermovement jump; SJ = Squat jump; Tcont = Average contact time; Tflight = Average flight time; Height = Average jump height; Power = Average jump power; Pace = Average jump pace; RSI = Average reactive force index; EI = Elasticity index.

**Table 3 ijerph-20-01539-t003:** Spearman’s rho correlations between HI, RPE, and the different CMJ and SJ components.

	HI	RPE
RPE	−0.345 ***	
Tcont_mean_CMJ	0.148 **	−0.090
Tflight_mean_CMJ	0.172 **	−0.118 *
Height_mean_CMJ	0.158 **	−0.149 **
Power_mean_CMJ	0.078	−0.119 *
Pace_mean_CMJ	−0.172 **	0.079
RSI_mean_CMJ	−0.091	−0.002
Tcont_mean_SJ	0.056	0.009
Tflight_mean_SJ	0.126 *	−0.026
Height_mean_SJ	0.113 *	−0.036
Power_mean_SJ	0.089	−0.024
Pace_mean_SJ	−0.063	−0.016
RSI_mean_SJ	−0.020	−0.018
EI_mean	0.022	−0.092

* *p* < 0.05; ** *p* < 0.01; *** *p* < 0.001; HI = Hooper Index; RPE = Perception of Effort; CMJ = Countermovement jump; SJ = Squat Jump; Tcont_mean = Average contact time; Tflight_mean = Average flight time; Height_mean = Average jump height; Power_mean = Average jump power; Pace_mean = Average jump pace; RSI_mean = Average Reactive Strength Index; EI = Elasticity Index.

**Table 4 ijerph-20-01539-t004:** Linear regression models explaining the HI and RPE.

	Hooper Index		RPE	
	B [95%CI]	*p*	B [95%CI]	*p*
Tcont_CMJ_mean	0.217 [0.047; 0.387]	0.013		
Height_CMJ_mean	0.060 [0.006; 0.113]	0.028		
EI_mean			−0.025 [−0.045; −0.006]	0.009
$R^{2}$	0.037	<0.001	0.016	0.009

B = Beta; *R*^2^ = proportion of variance; *p* = significance value; Tcont_mean = Average of contact time; Height_mean = Average of jump height; EI = Elasticity index; CMJ = Countermovement jump.

**Table 5 ijerph-20-01539-t005:** Linear regression models explaining the components of the CMJ and SJ.

	B [95%CI]	*p*
Tcont_CMJ_mean		
HI	0.107 [0.043; 0.172]	0.001
$R^{2}$	0.027	0.001
RPE		
$R^{2}$		
Tflight_CMJ_mean		
HI	0.003 [0.001; 0.005]	0.005
$R^{2}$	0.019	0.005
RPE	−0.003 [−0.005; 0.000]	0.023
$R^{2}$	0.012	0.023
Height_CMJ_mean		
HI	0.319 [0.114; 0.524]	0.002
$R^{2}$	0.023	0.002
RPE	−0.318 [−0.588; −0.048]	0.021
$R^{2}$	0.012	0.021
Power_CMJ_mean		
HI		
$R^{2}$		
RPE	−0.100 [−0.180; −0.020]	0.014
$R^{2}$	0.014	0.014
Pace_CMJ_mean		
HI	−0.011 [−0.019; −0.004]	0.003
$R^{2}$	0.021	0.003
RPE	0.012 [0.002; 0.021]	0.023
$R^{2}$	0.012	0.023
Tcont_SJ_mean		
HI	0.102 [0.028; 0.175]	0.007
$R^{2}$	0.018	0.007
RPE		
$R^{2}$		
Tflight_SJ_mean		
HI	0.002 [0.000; 0.003]	0.024
$R^{2}$	0.011	0.024
RPE		
$R^{2}$		
Height_SJ_mean		
HI	0.253 [0.062; 0.443]	0.010
*R* ^2^	0.016	0.010
RPE		
$R^{2}$		
EI_mean		
HI		
$R^{2}$		
RPE	−0.737 [−1.292; 0.183]	0.009
$R^{2}$	0.016	0.009

B = Beta; *R*^2^ = proportion of variance; *p* = significance value; Tcont_mean = Average on platform; Height_mean = Average of jump height; EI = Elasticity index; CMJ = Countermovement jump; HI = Hooper Index; RPE = Perception of Effort; SJ = Squat Jump; Tflight_mean = Average Flight Time; Power_mean = Average Jump Power; Pace_mean = Average Jump Pace; RSI_mean = Average Reactive Strength Index.

## Data Availability

The data presented in this study are available upon request from the corresponding author.

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
