# Peer review of "Relationship between Objective and Subjective Fatigue Monitoring Tests in Professional Soccer"

_ijerph, 2023, doi:10.3390/ijerph20021539_

Round 1

Reviewer 1 Report

I suggest some minor revisions that may improve the article.

Introduction

I consider this section to be too abbreviated, I recommend that you use references that you have used in the discussion to go deeper into the subject, as this way you can better understand the current situation of the subject in science and the previous research that has been carried out. 

I suggest that the authors incorporate the study hypothesis after the objective statement.

Materials and methods

The section is clear and very well presented. 

It could be interesting to differentiate between the results obtained between training sessions and matches.

Results

I do not understand the grouping of countries in table 1 on the description of nationalities.  It could be improved

Discusion

In line 258 of the conclusion the authors state "Also, it is important to note that perceived training effort responses and well-being scores depend on the type of task the players perform". I think I should explain the relationship between this statement and the task performed by the players analysed. In this sense, the results obtained in the variables after training and after matches could be related.

When the authors explain the 12 linear regression models found, I think you should give a more in-depth and substantiated explanation of the 12 models. Explain each one of them and try to give an explanation of the relationship, what is the reason for this, is this found in other research in football or another type of task? Personally I think it makes no sense to do the linear regression model and not give an individualised explanation of each model found.

I would like you to explain study 31 better, as the statement you make does not clarify well the relationship between pre-season additional km and fatigue.

Conclusions

The conclusions are appropriate, although I believe that I should elaborate on some relevant aspects of the 12 linear regression models found.

Reviewer 2 Report

General Assessment

Thank you for the opportunity to review this manuscript.

This study aimed to investigate the biopsychosocial aspect of fatigue during a sports season by studying the associations between objective and subjective fatigue tests in Professional Soccer.

Introduction

This section is well designed and well-written.

Line 80: Add an hypothesis.

Study design and Sample Size It needs to be written in more detail.

How was the selection of players (Sectoral position) and the number of players determined?

When and how were the measurements made?

How often were measurements made?

Was it measured only once? Or were measurements taken at intervals throughout the season?

How were the pre-test (measurement) loads determined? (game analysis or match analysis)

How were the results evaluated according to the players’ positions (Sectoral position)?

Round 2

Reviewer 2 Report

The article is eligible for publication.